# Observation of a high degree of stopping for laser-accelerated intense proton beams in dense ionized matter

Jieru Ren[1,11], Zhigang Deng[2,11], Wei Qi[2], Benzheng Chen[1,3], Bubo Ma[1], Xing Wang[1], Shuai Yin[1], Jianhua Feng[1], Wei Liu[1,4], Zhongfeng Xu[1], Dieter H. H. Hoffmann [1], Shaoyi Wang[2], Quanping Fan[2], Bo Cui[2], Shukai He[2], Zhurong Cao[2], Zongqing Zhao[2], Leifeng Cao[2], Yuqiu Gu [2], Shaoping Zhu[2,5,6], Rui Cheng[7], Xianming Zhou[1,8], Guoqing Xiao[7], Hongwei Zhao[7], Yihang Zhang[9,10], Zhe Zhang [9,10], Yutong Li[9,10], Dong Wu [3✉], Weimin Zhou [2✉] & Yongtao Zhao [1✉]

Intense particle beams generated from the interaction of ultrahigh intensity lasers with sample foils provide options in radiography, high-yield neutron sources, high-energy-density-matter generation, and ion fast ignition. An accurate understanding of beam transportation behavior in dense matter is crucial for all these applications. Here we report the experimental evidence on one order of magnitude enhancement of intense laser-accelerated proton beam stopping in dense ionized matter, in comparison with the current-widely used models describing individual ion stopping in matter. Supported by particle-in-cell (PIC) simulations, we attribute the enhancement to the strong decelerating electric field approaching 1 GV/m that can be created by the beam-driven return current. This collective effect plays the dominant role in the stopping of laser-accelerated intense proton beams in dense ionized matter. This finding is essential for the optimum design of ion driven fast ignition and inertial confinement fusion.

[1] MOE Key Laboratory for Nonequilibrium Synthesis and Modulation of Condensed Matter, School of Physics, Xi'an Jiaotong University, Xi'an 710049, China. [2] Science and Technology on Plasma Physics Laboratory, Laser Fusion Research Center, China Academy of Engineering Physics, Mianyang 621900, China. [3] Institute for Fusion Theory and Simulation, Department of Physics, Zhejiang University, Hangzhou 310058, China. [4] Xi'an Technological University, Xi'an 710021, China. [5] Institute of Applied Physics and Computational Mathematics, Beijing 100094, China. [6] Graduate School, China Academy of Engineering Physics, Beijing 100088, China. [7] Institute of Modern Physics, Chinese Academy of Sciences, Lanzhou 710049, China. [8] Xianyang Normal University, Xianyang 712000, China. [9] Beijing National Laboratory for Condensed Matter Physics, Institute of Physics, Chinese Academy of Sciences, Beijing 100190, China. [10] School of Physical Sciences, University of Chinese Academy of Sciences, Beijing 100049, China. [11] These authors contributed equally: Jieru Ren, Zhigang Deng. ✉email: dwu.phys@zju.edu.cn; zhouwm@caep.cn; zhaoyongtao@xjtu.edu.cn

Alpha-particle stopping in dense ionized matter is essential to achieve ignition in inertial confinement fusion[1–5]. Fast ignition (FI) relies even more on a detailed understanding of ultrahigh-current ion stopping in matter, which is therefore considered as a fundamental process of utmost importance to nuclear fusion. In the fast ignition scheme[6–9], a short and intense pulse of energetic charged particles—electrons, protons, or heavy ions—generated by an ultra-high-intensity laser, is directed toward the pre-compressed fusion pellet. The charged-particle beam requirements to achieve ignition have been discussed and studied in detail previously[10–14] based on single-particle stopping theory. However, the collective effects induced by high-current charged-particle beams could alter significantly the projected range, the magnitude of energy deposition, and therefore change the requirements for ignition correspondingly. Besides, in the cases of ion beam-driven inertial confinement fusion and high-energy density science, which require ultrahigh beam intensity from accelerators[15–19], no collective effects on ion stopping processes due to high beam intensity are considered nor—to the best of our knowledge—were they reported in any previous experiments.

Since the discovery of alpha decay and the availability of energetic fission fragments, it became interesting to study fast particle stopping processes in matter[20–25]. In past decades, numerous theoretical models[26–32], some of which can be considered to be further developments of the early work of Bethe[28] and Bloch[29], are built to describe individual charged particle stopping in dense ionized matter. Only recently experiments with sufficient precision were carried out with dense ionzied matter to distinguish between different models[25,33–35]. In these experiments, incident particles are generated from laser-induced nuclear reactions[33,34], or from traditional accelerators[25,35]. Hence the beam intensity was low, and the individual ion stopping theories can be discriminated[36].

Ultra-high-intensity lasers ($10^{18}$–$10^{22}$ W/cm$^2$) have opened up perspectives in many fields of research and application[37–44]. By irradiating a thin foil with ultra-high-intensity lasers, an ultrahigh accelerating field (1 TV/m) can be formed and multi-MeV ions with high intensity ($10^{10}$ A/cm$^2$) in short timescale (~ps) are produced[45–53]. Such beams provide experimental opportunities to investigate the beam-driven complex collective phenomena[54–59]. In particular, the stopping power for these intense beam could be orders of magnitude higher than that for individual particles if the beam intensity is high enough[60–63]. In our previous experiment, we sent the laser-accelerated ion beams directly into the plasma target, and observed that the energy spectra of the ions were significantly downshifted after passing though the dense plasma[64]. This energy downshift was far beyond the Bethe–Bloch predictions (see Supplementary Figs. 8 and 9 for details). However, the large energy spread of the incident beam makes it difficult to correctly interpret the results.

In this article, we improve the precision of the measurement by using a magnetic dipole to trim out a quasi-monoenergetic proton beam. The dense ionized target is produced by irradiating a tri-cellulose acetate (TCA) foam sample with soft X-rays from a laser-heated hohlraum. Thus the temperature and density are homogeneous across the ionized target. This state can be maintained for a period of more than 10 ns. This period is two to three orders of magnitude longer than the beam duration and the beam-plasma interaction timescale. Therefore, the target can be considered to be quasi-static. This kind of experimental scenario allows for precise measurement of intense proton beam stopping in dense ionized matter. We observed that the energy loss is enhanced by one order of magnitude in comparison to the predictions from individual-proton stopping theories, Bethe–Bloch[28,29], Li–Petrasso (LP)[26], standard stopping model (SSM)[32]. Through PIC simulation, we attribute the high degree of enhancement to a strong decelerating electric field induced by the intense proton beam. This collective effect is the primary cause for the enhanced stopping, and it is likely to have a major impact on nuclear fusion scenarios like fast ignition, alpha-particle self-heating, as well as ion driven inertial confinement fusion.

## Results

The experiment was carried out at the XG-III laser facility of Laser Fusion Research Center in Mianyang. The experimental layout is displayed in Fig. 1. Here a short and intense laser beam of 800 fs duration, 20 μm focal spot, and 150 J total energy irradiates a CH-coated tungsten foil (15-μm thick) to generate a charged-particle beam. The beam consists of a mixture of protons (H$^{1+}$) and carbon ions with different charge states (C$^{1+}$, C$^{2+}$, C$^{3+}$, and C$^{4+}$). They originate at the backside of the target by means of the target normal sheath acceleration (TNSA). The predominant particle species is H$^{1+}$, because the charge- to mass-ratio is maximum for this species, and is, therefore, more effectively accelerated than the lower charge-to-mass ratio species of carbon ions. The TNSA mechanism results in a broad range of particle energies, which is not favorable for quantitative analysis of the particle energy loss. A magnetic dipole, with entrance and exit slits, was used to generate a monoenergetic beam. The ions, spatially collimated by the 500 μm entrance slit, are dispersed laterally by the magnetic dipole according to their specific $\mathbf{p}/q$ value, where $\mathbf{p}$ and $q$ are the particle momentum and charge, respectively. A second 500 μm exit slit, selects the quasi-monoenergetic ion pulses. The selected ions consist of different particle species, with similar $\mathbf{p}/q$ value, they have, however, different velocities and therefore arrive at the target pulse by pulse with different time delay. In the current case, the C$^{4+}$ ion pulse lags behind proton pulse by about 30 ns at the plasma target, and C$^{3+}$, C$^{2+}$, and C$^{1+}$ pulses are delayed more, therefore the laser-accelerated carbon ions have no influence on the proton beam stopping measurement.

A gold hohlraum converter was used to generate soft X-rays by interaction of a ns laser pulse (150 J), with the hohlraum walls. The X-rays subsequently irradiated and heated the foam target (C$_9$H$_{16}$O$_8$, density of 2 mg/cm$^3$ and thickness of 1 mm) to produce ionized matter. The hydrodynamic response of the heated foam target under this kind of scheme has been very well-investigated and the state has been well characterized[65–67]. Once heated by soft X-rays from hohlraum radiation, the material expansion occurs inside the target between the sponge-like structures. This micro expansion leads to target homogenization, while the volume and density of the entire target stay constant for more than 10 ns. Therefore, homogeneous, ns-long-living, and quasi-static ionized matter is generated. Unlike producing ionized matter through direct heating by high-power lasers—in which case strong electromagnetic fields are generated in the target and will greatly influence the proton transportation behavior[68–70]—here the electromagnetic fields resulted from the heating process can be neglected.

In order to determine plasma parameters, the emission spectra of the gold hohlraum and target matter were measured. The gold hohlraum radiation spectrum is well-represented by a 20 eV black body radiation spectrum, while the temperature of the plasma target is 17 eV. This value was obtained from a Boltzmann slope analysis of the He-like carbon lines. Given a temperature of 17 eV, and mass density of 2 mg/cm$^3$, the number density of free electrons is determined to be $4 \times 10^{20}$ cm$^{-3}$ based on the FLYCHK code[71].

A Thomson parabola spectrometer (TPS)[72–74] in conjunction with a plastic track detector CR39 was used to obtain the energy spectrum of the charged particles. The energy resolution of the

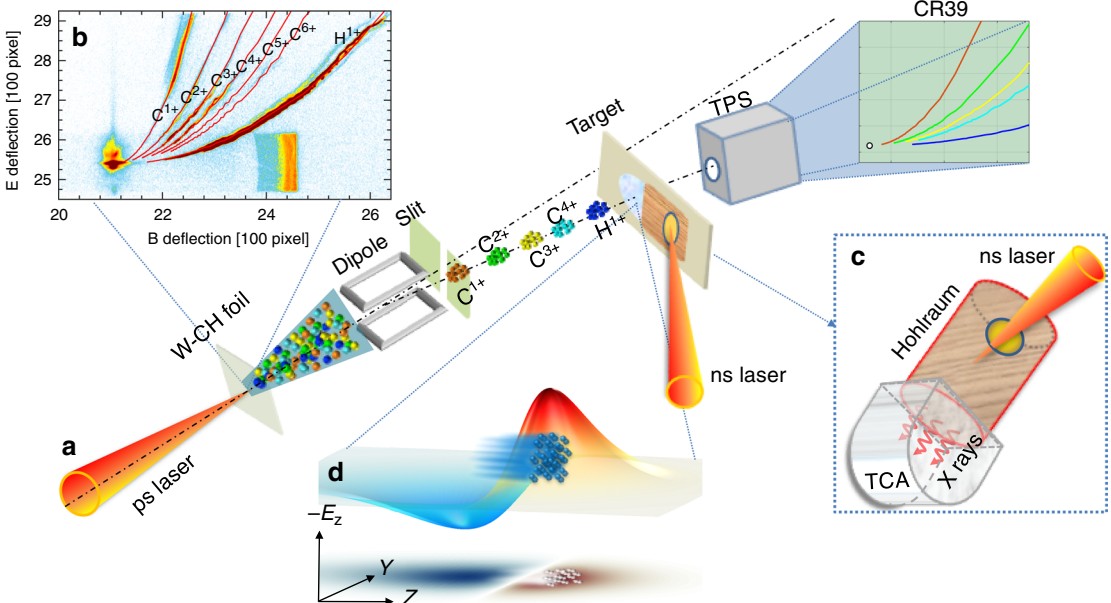

**Fig. 1 Layout of the experiment. a** A ps laser is focused onto a tungsten foil, generating intense short-pulse ion beams with different species. A magnetic dipole with slits at the entrance and exit serve as $p/q$ analyser to select monoenergetic ion beams. Such ions interact with the laser-generated plasma target and emerge from the target with a lower energy due to the incurred energy loss. The final-state energy is measured by a Thomson parabola in conjunction with CR39 film. **b** Parabola spectra of laser-accelerated ions without dipole measured by Thomson parabola in conjunction with Fuji image plate. **c** The target consists of a gold hohlraum converter to produce the soft X-rays that irradiate the TCA foam to generate a dense ionized sample. **d** The insert shows the simulation result of an intense proton beam moving along the z direction, inducing a strong longitudinal electric field, which is counter-directional to the proton beam propagation, causing the unusual high degree of stopping.

TPS achieves $E/\delta E \sim 34$ for protons at energy $E = 3.36$ MeV (see section I of the Supplementary information for details), where $\delta E$ is the energy range covered by the incident beam spot on the detector. In Fig. 2a, tracks recorded on CR39 film are displayed for ions passing through the system with/without target. When the plasma target is inserted, only protons are observed in the TPS. The deflection distances of protons without/with target are converted to energies in Fig. 2b and c, respectively. The energy distribution of the incident, unperturbed protons without target centers at 3.36 MeV, and the full width at half maximum (FWHM) is about 0.10 MeV. After passing through the plasma target, the central energy is downshifted to 2.98 MeV and the FWHM increased to about 0.25 MeV.

## Discussion

The energy loss of intense ion beam in ionized matter is composed of two terms as $dE/dx = (dE/dx)_{\text{collision}} + (dE/dx)_{\text{collective}}$. The first term $(dE/dx)_{\text{collision}}$ describes the collisional stopping induced by binary interaction of the individual projectiles with the individual particles in the plasma. The second term $(dE/dx)_{\text{collective}}$ describes the collective stopping induced by the beam-driven electric fields. $(dE/dx)_{\text{collision}}$ consists of contributions from free electrons and plasma ions. The ionic contributions for partially ionized plasma include two parts, bound electrons and nuclei. In the current regime, where the protons are much faster than the thermal electrons, the contribution of the nuclei to $(dE/dx)_{\text{collision}}$ can be neglected[75–77]. Here in this article the nuclear stopping is excluded.

In Fig. 3, the measured energy loss is compared to theoretical models, e.g., Bethe–Bloch model, Li–Petrasso (LP) theory, and SSM by Deutsch. These theories are based on binary collisions with free electrons, bound electrons, and/or plasmons. They all underestimate the measured stopping power by as much as one order of magnitude, even when considering an error of about 15%

from the uncertainty of plasma electron density. We attribute this unusual high degree of stopping to collective electromagnetic effects induced by high-current ion beams.

In order to understand this enhanced stopping, both collective electromagnetic effects and close particle–particle interactions need to be taken into account. The most appropriate tool to simulate the conditions of the experiment is the particle-in-cell method (PIC), which in recent years has established itself as a state-of-the-art method for solving problems of kinetic plasma physics[54,78,79].

We used the recently developed PIC code LAPINS[79,80], which is able to simulate intense beam-plasma interaction in a self-consistent way, which contains both close collisions and collective electromagnetic fields (see details in "Methods"). The simulation assumes, the incident proton beam to have Gaussian distribution in space and time, with a beam duration of 1 ps and a transverse extension of 1 mm. The energy spectrum is also assumed to be Gaussian, with the peak of the energy distribution at 3.36 MeV and FWHM of 0.10 MeV. The measured ionized target parameters were used as simulation input. The simulation was carried out in $Z–Y$ Cartesian geometry with beam propagating along the $Z$ direction. The size of the simulation box was 1.2 mm × 2.5 mm, with a grid size of 0.75 μm × 25 μm.

Given the incident proton beam with density of $8 \times 10^{16}$ cm$^{-3}$, which corresponds to high-current case of $3 \times 10^7$ A/cm$^2$, Fig. 4a shows the longitudinal electric field $E_z$ induced by the beam-driven return current after a propagation distance of about 0.3 mm. A strong decelerating field approaching $10^9$ V/m is generated, and contributes to the proton stopping. The proton energy spectrum after passing through 1 mm of plasma is shown in Fig. 4c. The energy spread is significantly broadened compared to the initial spread. We attribute this to a decreasing field, the protons are imbedded in. Protons with higher energies are located at the front end of bunch and therefore experience a smaller

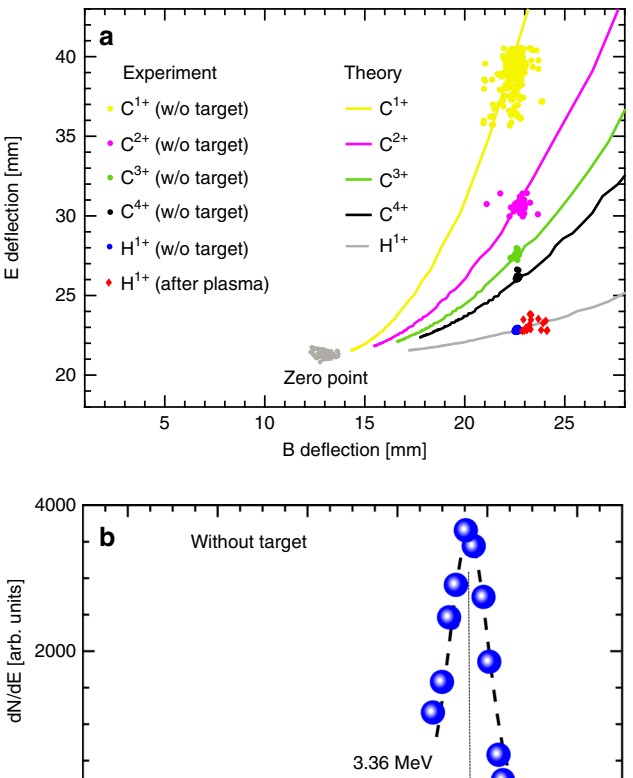

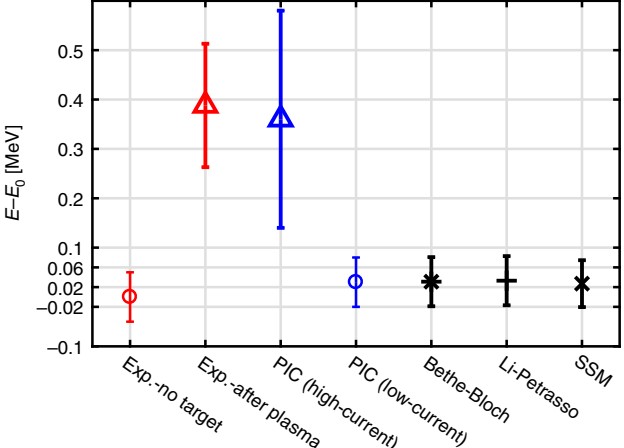

**Fig. 3 Measured (exp.-after plasma), numerical (PIC), and analytical predictions (Bethe–Bloch, SSM, and Li–Petrasso) of the proton beam energy spectra downshift after passing through the dense plasma target in relative to the central energy $E_0$ of incident proton beam.** The symbols represent the central energies and the bars represent the FWHMs of the respective spectra. For comparison, the data for the incident proton beam (no target) are shown as well.

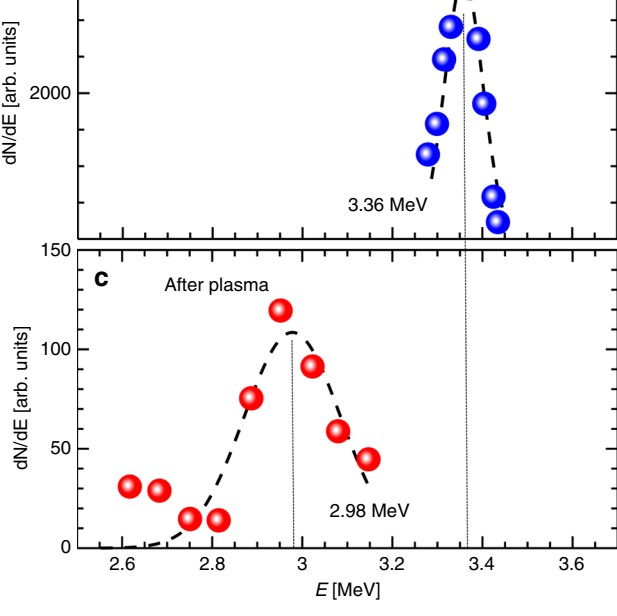

**Fig. 2 TPS CR39 tracks of quasi-monoenergetic ions passing through the system with/without target and the converted energies of protons. a** Tracks of ions recorded on TPS CR39 passing through null target and plasma target. The $X$ and $Y$ coordinates represent the magnetic and electric deflection distances. The dots with the same B deflection distance from up to down (yellow, magenta, green, black, and blue in order) represent tracks formed by $C^{1+}$, $C^{2+}$, $C^{3+}$, $C^{4+}$, and protons passing through the system without (w/o) target. Tracks for protons passing through the plasma are represented by red dots. The tracks for the zero reference point are indicated with gray dots. The solid curves represent the theoretical tracks of these ion species with various energies. **b** Energy spectra of the protons passing the system without target. The experimental distribution (blue dots) is well fitted by a Gaussian profile (dashed line) with central energy of 3.36 MeV (dotted line). **c** Energy spectra of the protons passing the system without target. The experimental distribution (blue dots) is fitted by a Gaussian profile (dashed line) with central energy of 2.98 MeV.

decelerating electric field than those with lower energies that come later. The spatial size of this decelerating field is comparable to the size of the proton bunch. This is different from the plasma wakefield case[81,82], where the spatial structure of the electric field is determined by the plasma density. Here the plasma wakefield

wavelength is much smaller than the beam length, therefore the wakefield-induced collective acceleration and deceleration cancel out. The central energy of proton spectrum is downshifted by 0.36 MeV after passing through the plasma. As shown in Fig. 3, this energy shift (blue triangle) agrees with experimental data in magnitude. We carried out additional simulations for different beam densities at $8 \times 10^{11}$ cm$^{-3}$ and $8 \times 10^{15}$ cm$^{-3}$, which are defined as low- and intermediate-current cases, respectively. For the low-current case, the beam-induced longitudinal electric field $E_z$ after propagating for 0.3 mm in the plasma is shown in Fig. 4b. No collective decelerating field is excited under such conditions. After passing through the plasma, the energy spectrum is downshifted by only 0.03 MeV as shown in Fig. 4c. This prediction agrees well with those calculated by the different binary collision theories, as shown in Fig. 3, which indicates the dominant role of collisional stopping in low-current cases. As for the intermediate case, the stopping due to the collective effects is comparable to that caused by binary collisions, giving rise to an energy loss of 0.06 MeV as shown in Fig. 4c.

In all, the energy loss of laser-accelerated intense proton beam in dense ionized matter consists of $(dE/dx)_{collision}$ and $(dE/dx)_{collective}$. Bethe–Bloch, LP, SSM models, and PIC simulation for low-current case give similar predictions for $(dE/dx)_{collision}$, which is one order of magnitude lower than the experimental data. PIC simulation for high-current cases shows that when sending a very dense ion bunch into the plasma, strong electric fields can be induced, and the ion bunch is imbedded in the deceleration field. This leads to a significant enhancement of the energy loss, which fairly well explains our observation.

In summary, the laser-accelerated intense proton beam stopping in a dense ionized matter has been measured. Benefiting from the fact that we have a quasi-monoenergetic proton beam and long-living well-characterized dense ionized target, accurate stopping power data were obtained. The measured stopping power exceeds the classical theory predictions in binary collision scheme by about one order of magnitude. The phenomenon can be very well explained by our PIC simulation combined with a Monte Carlo binary collision model and a reduced model taking account of the collective electromagnetic effects. The stopping power is

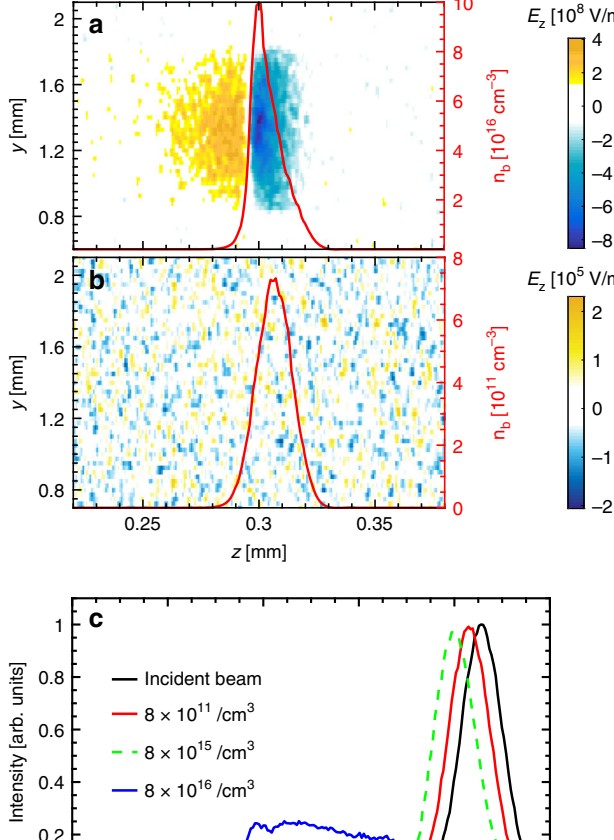

**Fig. 4 Simulated distribution of longitudinal electric field and proton beam density during beam transport in experimentally used ionized target, and the resulting proton beam energy spectra shift after passing through the sample. a** Longitudinal electric field driven by proton beam, moving along the $Z$ direction with a beam density of $8 \times 10^{16}$ cm$^{-3}$. The beam density profile is indicated by the red solid curve. **b** The same situation as in (**a**) but the initial beam density is reduced to $8 \times 10^{11}$ cm$^{-3}$. **c** Normalized energy spectra. The energy spectra of protons without target are represented by the black solid curve. The colored lines indicate the energy spectra of protons passing through the plasma target for varying beam densities of $8 \times 10^{11}$ cm$^{-3}$ (red solid line), $8 \times 10^{15}$ cm$^{-3}$ (green dashed line), and $8 \times 10^{16}$ cm$^{-3}$ (blue solid line).

dramatically enhanced due to the return-current-induced decelerating electric field approaching 1 GV/m. We have demonstrated the existence of collective effects, for high-density beam, leading to enhanced stopping. This will be important for the optimum design of ion driven inertial confinement fusion and fast ignition scenarios.

## Methods
The collisional model in the current PIC code is based on Monte Carlo binary collisions, which has the advantage of calculating the beam stopping in a natural manner. The model includes binary collisions among electron–electron, electron–ion, and ion–ion, taking into account contributions from both free and bound electrons. Compared with other existing models, physical quantities, such as angular scattering, momentum transferring, and temperature variation, can be taken into account quite readily in the approach.

In the calculations, three steps are made iteratively: (i) pair of particles are selected randomly in the cell, i.e., either electron–electron, electron–ion, or ion–ion pairs; (ii) for these pair of particles, the binary collisions are associated with changes in the velocity of the particles within the time interval $\delta t$, which are

calculated; (iii) and then the velocity of each particle is replaced by the newly calculated one.

In order to contain both bound and free electrons' contribution into the binary collision model, we here take the collision frequency between ions and electrons, in the above (ii) step, as,

$$\nu_{\text{i}-\text{e}} = \frac{8\sqrt{2\pi}e^4 Z_b^2 Z n_i}{3m_e^2\beta^3}[\ln(\Lambda_{\text{f}}) + \frac{A-Z}{Z}\ln(\Lambda_{\text{b}})], \quad (1)$$

where

$$\ln(\Lambda_{\text{b}}) \equiv \ln[\frac{2\gamma^2 m_e\beta^2}{\bar{I}_A(Z)}] - \beta^2 - C_K/A - \delta/2, \quad (2)$$

and

$$\ln(\Lambda_{\text{f}}) \equiv \ln(\lambda_D/b). \quad (3)$$

$A$ is the atomic number of stopping medium, $Z$ is the ionization degree of background plasmas, $n_i$ is the nucleus density of stopping medium, $m_e$ is the electron mass, $\gamma$ is the relativistic factor of the projected ions, $\beta$ is the velocity of projected ions, $\bar{I}_A$ is the average ionization potential, and $Z_b$ is the effective charge state of injected ion beams, which equals '1' for the case of protons in our present studies. In Eq. (1), the latter two terms are the shell correction term and the density effect correction term, respectively. These two terms are based on Fano's original work[83], to which the definitions of $C_K/A$ and $\delta/2$ can be referred. The Debye length, $\lambda_D$, is a dynamic value changing as $\lambda_D = \sqrt{(T_e/4\pi n_e)(1+\beta^2/v_{\text{th}}^2)}$, where $T_e$ and $v_{\text{th}}$ are the temperature and thermal velocity of background electrons. Parameter $b$ is the distance of closest approach between the two charges. Especially, $(A-Z)/Z$ defines the ratio of bound electrons' contributions. For fully ionized plasmas, $Z \to A$, the collision frequency between ions and electrons converges to

$$\nu_{\text{i}-\text{e}} \sim \frac{8\sqrt{2\pi}Z_b^2 e^4 Z n_i}{3m_e^2\beta^3}\ln(\Lambda_{\text{f}}). \quad (4)$$

For neutral atoms, $Z \to 0$, in contrast, the frequency is

$$\nu_{\text{i}-\text{e}} \sim \frac{8\sqrt{2\pi}Z_b^2 e^4 A n_i}{3m_e^2\beta^3}\ln(\Lambda_{\text{b}}). \quad (5)$$

At the low-temperature limit, when all electrons are bound at the nucleus, the calculated stopping powers converge to the NIST ones with the average ionization degree approaching zero as the stopping powers of cold materials can be well calculated by Bethe–Bloch formula. With the increase of temperature, more and more bound electrons are ionized, giving rise to an increased stopping power to cold matter. When the temperature is further increased, with ionizations reaching the maximum, lowered stopping power is observed, which is due to the suppression of collision frequency between projected proton beam and hot plasmas in the target.

Simulation of large scale plasmas often results in an intractable burden on computer power. Therefore, instead of solving the full Maxwell's equations, we combine the PIC method with a reduced model[79]. To take into account collective electromagnetic effects, the background electron inertia is neglected, and instead the background return current is evaluated by the Ampere's law $\mathbf{J}_e = (1/2\pi)\nabla\times\mathbf{B} - (1/2\pi)\partial\mathbf{E}/\partial t - \mathbf{J}_b - \mathbf{J}_i$, where $\mathbf{B}$ is the magnetic field, $\mathbf{E}$ is the electric field, $\mathbf{J}_b$ is the injected beam current, and $\mathbf{J}_i$ is the background ion current. Applying the continuity equation $\nabla\cdot\mathbf{J} + \partial\rho/\partial t = 0$ with the total current $\mathbf{J} = \mathbf{J}_b + \mathbf{J}_i + \mathbf{J}_e$, where $\rho$ is the charge density, the Poisson equation $\nabla\cdot\mathbf{E} = 2\pi\rho$ is rigorously satisfied. The electric fields are then obtained from Ohm's law, $\mathbf{E} = \eta\mathbf{J}_e - \mathbf{v}_e\times\mathbf{B}$, where $\mathbf{v}_e$ is the background electron velocity, and $\eta$ is the resistivity. Taking advantage of the Monte Carlo collision model, resistivity $\eta$ is obtained by averaging over all binary collisions at each time step for each simulation cell. Finally, Faraday's law is used to obtain the magnetic fields $\partial\mathbf{B}/\partial t = -\nabla\times\mathbf{E}$. This field solver, which couples Ampere's law, Faraday's law, and Ohm's law, can completely remove the numerical heating and reduces significantly the numerical expense. With these advantageous features a unique tool is at hand, which can self-consistently model transport and energy deposition of intense charged particles in dense ionized matter.

## Data availability
The datasets generated and analyzed during the current study are available from the corresponding authors upon reasonable request. The simulation details are available from the corresponding author on reasonable request.

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

## Acknowledgements

We sincerely thank Olga Rosmej from GSI Helmholtzzentrum für Schwerionenforschung for the physical discussion, as well as the staff from Laser Fusion Research Center, Mianyang for the laser system running and target fabrication. Besides, we want to thank Wei Kang and Jingli Gao from Peking University for their attempt in calculations with TDDFT (time-dependent density functional theory coupled to Ehrenfest dynamics) calculation. Part of the simulations are performed at Qilin-2 supercomputer at Zhejiang University. The work is supported by Chinese Science Challenge Project No. TZ2016005, National Key R & D Program of China No. 2019YFA0404900, National Natural Science Foundation of China (Grant Numbers U2030104, 11705141, 11775282, and U1532263), China Postdoctoral Science Foundation (Grant Numbers 2017M623145 and 2018M643613), and the Fundamental Research Funds for the Central Universities.

## Author contributions

Yongtao Zhao conceived this work, organized the experiments and simulations with Weimin Zhou and Dong Wu, respectively. Jieru Ren, Zhigang Deng, and Yongtao Zhao carried out the experiment together with the high-power laser team (Zongqing Zhao, Weimin Zhou, and Yuqiu Gu), the plasma diagnostics team (Shaoyi Wang, Quanping Fan, Bubo Ma, Bo Cui, Xing Wang, Zhurong Cao, and Leifeng Cao), ion beam characterization team (Wei Qi, Shuai Yin, Shukai He, Wei Liu, Rui Cheng, Xianming Zhou, Jianhua Feng, Yihang Zhang, and Zhe Zhang). Jieru Ren, Zhigang Deng, Shuai Yin, Wei Qi, Shaoyi Wang, and Quanping Fan analyzed the main part of the experimental data. Benzheng Chen, Jieru Ren, Yongtao Zhao, and Dong Wu performed the simulations and related analysis. Shaoping Zhu, Zhongfeng Xu, Guoqing Xiao, Hongwei Zhao, Yutong Li, Yuqiu Gu, and Leifeng Cao contribute in the physical discussion. Jieru Ren, Yongtao Zhao, Dong Wu, and Dieter H.H. Hoffmann wrote the paper.

## Competing interests

The authors declare no competing interests.
