## [Peer Review File · Nature Communications]

Reviewers' comments:

Reviewer #1 (Remarks to the Author):

The authors advocate an order of magnitude increase for the stopping of MeV protons in a CHO foam target ns-laser irradiated into a 17 eV and $N_e=10^{20}/\text{cc}$ classical electron plasma. The present laser protocol of laser-produced protons stopped synchronously into a laser created plasma target obviously elaborates on the scheme already given in ref.49 with the LLNL Titan laser.

The supplemental sections are highly detailed and instructive.

To be definitively convinced of the soundness of the authors assessment, casual readers are still in need of a proof that

-there is no mix of highly charged Carbon ions with the velocity selected protons

-that the highly charged target C and O ions do not contribute to the MeV proton stopping despite that proton velocity is above the target thermal electron one.

If the authors could achieve that goal, they would add a splendid contribution to the very challenging field of proton stopping in dense classical plasma.

Reviewer #2 (Remarks to the Author):

begin_report:

Major comments and report summary

In their manuscript "Anomalous stopping of laser-accelerated intense proton beam in dense ionized matter" the authors present results both collected in an experiment and modeling using the PIC simulation technique on stopping of protons in an isochorically heated target medium.

I do not recommend publication of this manuscript in its current form for two major reasons given in the following.

While the presented scientific results on stopping power seem sound and are indeed interesting, (1) the style of presentation is poor across the manuscript (see comments and highlights in the annotated PDF) and (2) I am not convinced the theoretical methods used to compare against experiment are sufficiently accurate in order to interpret the observed enhancement in stopping as "anomalous".

In order to increase the impact of their current work, a supplemental theoretical treatment within a more accurate method (such as time-dependent density functional theory coupled to Ehrenfest dynamics) would be required to warrant publication as an article in Nature Communications. The current scientific results, however, would be suitable for publication as an article in Scientific Reports or a similar journal.

Summary of detailed comments (given in the annotated PDF)

Page 1, left:

the highest beam intensity from accelerators

Style: what do you mean with "the highest beam intensity"? Elaborate.

Page 1, right:

We ever carried out an experiment in previous, and observed a significant enhancement of energy loss for the laser-accelerated intense proton beam (see [58] and supplementary material for details).

Style: this sentence is very unclear, rewrite.

Page 2, left:

However, it was di

cult to conduct quantitative analysis due to the large energy spread of the beam. In order to improve our understanding of these effects, experimental data with high precision are required.

Style: unclear, rewrite.

I believe, the motivation of this work is a previously conducted experiment Ref. 58. Please rewrite this paragraph accordingly.

Page 2, left:

The hydrodynamic timescale of the target is long compared to the proton beam pulse, hence the target can be considered to be quasi static with well-characterized parameters.

please be more quantitative: give explicit estimates for the timescales in order to justify a quasi-static approximation.

Page 2, left:

We observed enhanced stopping by one order of magnitude compared to the classical single particle slowingdown theory.

This seems to be the central result of this work. However, the style of presentation in this paragraph is very poor.

Also, a classical single-particle theory is used to compare against the experimental result.

However, it is not clear at all which theory the authors compare against. Do you compare against (1) Bethe-Bloch model, (2) Li-Petrasso (LP) theory or something else? Please clarify and add citations accordingly.

Page 2, left:

We attribute this phenomenon to a strong decelerating electric field induced by the intense proton beam.

Refer to results and figures here.

Page 3, right:

Due to the large penetration of soft x-rays, the foam was heated quasi isochorically. Clarify: isochorically or quasi-isochorically? If quasi-isochorically, please explain.

Page 3, right:

In Fig. 3, the measured energy loss is compared to theoretical models, e.g. Bethe-Bloch model, Li-Petrasso (LP) theory [38] and Standard Stopping Model (SSM) by Deutsch [44]. These theories are based on binary collisions with free electrons, bound electrons and/or plasmons. They all underestimate the measured stopping power by as much as one order of magnitude. We therefore call the observed effect anomalous stopping and attribute this to collective electromagnetic effects induced by high-current ion beams.

Fair enough, but these models are very simple. In my opinion, a comparison with very simplified theoretical models is not sufficient to conclude that the experimentally observed stopping is anomalous. A comparison with a higher level of theory is required. The observed result is still interesting, but I believe, a more thorough theoretical analysis is required to attribute an "anomaly" to the observed stopping. I would suggest to compare against results computed within time-dependent density functional theory coupled to Ehrenfest dynamics which is the current "gold standard". These calculations will be computationally expensive but doable under the reported conditions (a temperature of 17 eV, mass density 2 mg/cc, MeV velocity of projectile).

Page 3, right:

In order to understand this anomalous stopping, both collective electromagnetic effects and close particle...

Bad style of presentation: Only here it becomes clear that the authors actually perform additional modeling in order to find an interpretation of the unusually high degree of stopping observed. This was not mentioned at all up to this point. I recommend the authors rewrite the abstract and their introduction and clearly explain their methodology, particularly with regard to the PIC simulations.

Page 4, left, Fig. 3:

Can you also give error estimates for the models (Bethe-Bloch, Li-Petrasso, SSM)?

Page 4, left:

The most appropriate tool to simulate the conditions of the experiment is the Particle-in-Cell method (PIC), which in recent years has established itself as a state-of-the-art method for solving problems of kinetic plasma physics.

Add citation(s) to seminal paper(s) and point to supplemental material for details on the theory used.

Page 5, right:

We used the newly developed PIC code LAPINS [65, 66], where...

The authors should have referenced these citations already when they discuss their PIC calculations. Although references are given, the authors should also provide a concise and clear description of their PIC methodology either here or in the supplementary material.

end_report

Reviewer #3 (Remarks to the Author):

Comments on "Anomalous stopping of laser-accelerated intense proton beam in dense ionized matter" written by Jieru Ren et al., to be submitted to Nature Communications. In this paper, the authors mention experimental results of anomalous stopping power of MeV ion beam in dense ionized matter. They use laser generated ion beam with magnetic dipole energy filter and radiation heated target with high energy laser pulse. The method itself looks interesting and their idea to generate quasi-monochromatic ion beam from very broad TNSA ion beam looks well organized operation. They claim that they measured anomalous stopping phenomena with energy shifted ion beam energy and comparison with the PIC simulation prediction. The reviewer carefully read this manuscript and it is several difficulties to understand the details of the experiments and to judge their claims. The reviewer lists up the uncertain points and considers the authors should make them clear in their manuscript.

(1) There is no information about the current of quasi-monochromatic ion beam. That ion beam is sliced with single magnetic dipole filter. Therefore, the current itself largely decreases from the generation point. In addition, energy angle dispersion of ion beam remains after passing through the dipole magnet so that the cross section of ion beams gradually increases after the slit of the magnetic dipole filter. In that situation, how much current density can they expect and is that number still high enough to use the quasi-monochromatic ion beam for such scientific investigation? They need to show it.

(2) In such energy loss investigation, comparison between perturbed and unperturbed ion spectrum is an essential part. However, there is almost no information about how to get this unperturbed ion spectrum. They only said "without target". This means nothing between quasi-monochromatic filter and the spectrometer. If that so, it is some difficulties to use these different transport circumstance condition results as reference. If they have heated and unheated target results, that would be more reliable.

(3) In many TNSA ion beam probe experiments, there are transit potential effects of the high current ion beam illuminated target. There are many distorted mesh pictures from the ion beam deflection but there are negligible small electrostatic potential effects of the target. (for example, Mackinnon et al., Rev. Sci. Instrum., vol.75, No.10, p.3531 (2004)) They need to mention about their loss of energy purely occurring inside the heated target.

(4) Normally, Thomson parabola spectrometer needs a pinhole entrance pinhole to define the starting point before deflection of electric field and magnetic field. But, this pinhole plays a role to produce additional potential effects if the ion current density is so high. In their experimental setup, there is no information about it. In addition, in Fig. 2(a), there is a relatively broad "zero point" signal. It may be neutral components of charge exchange process. But that narrowness of neutral component in the Thomson spectrometer is one of the experimental evidences of the energy resolution. It looks large width and spreading in the only B field direction. To clarify the energy difference in their experiments, they need to clarify their spectrometer condition and resolution.

more seriously.

(5) Actually, there are many papers related to the anomalous stopping of ion beams. They need to add more references. Specially, they miss old Sandia's ion beam fusion experiments. (for example, F. C. Young, PRL, Vol. 49, No.8, p.549 (1982), E. J. McGuire et al., Phys. Rev. A, Vol. 26 No.3, p.1318 (1982))

(6) In the previous accelerated ion beam experiment, there are ionizing effect of the target is one of the mechanism to explain enhancement stopping power. In the author's experimental condition, target carbon atoms are not fully ionized. Therefore, they need to include this effect.

(7) The author's main claiming points is induced electric field affecting to the stopping power of ion beams. If they want to appeal this, the reviewer consider the comparison between low current density experiment and high current density experiment. But they have no data about that. In this case, it is very difficult to say their mechanism is dominated process. If they have some other experimental data related and can be used for explanation of their model, they need to show them.

Reply to the Referee #1:

Remark 1: *The authors advocate an order of magnitude increase for the stopping of MeV protons in a CHO foam target ns-laser irradiated into a 17 eV and $N_e = 10^{20}$ /cc classical electron plasma. The present laser protocole of laser-produced protons stopped synchronously into a laser created plasma target obviously elaborates on the scheme already given in ref. 49 with the LLNL Titan laser. The supplemental sections are highly detailed and instructive.*

Response: We thank the referee for appreciating our work.

Remark 2: *To be definitively convinced of the soundness of the authors assessment, casual readers are still in need of a proof that*

1) there is no mix of highly charged Carbon ions with the velocity selected protons.

Response and revision: It is for sure that there is no mix of highly charged carbon ions with the velocity selected protons. The reason is as follows.

The selected ions have the same $m\mathbf{v}/q$ value, where m is the ion mass, \mathbf{v} is the ion velocity, and q is the charge state of the ion. Therefore, different ion species have different velocities, so that different ion species will arrive at the target with different time delays, as is shown in Table I, which is in reference to protons. It can be seen that the carbon ions lag behind the protons by tens to hundreds of nanoseconds.

In the revised manuscript, we further underline this process (see details in the first paragraph of section "experimental setup and data taking"). Besides, in the supplementary material, the velocities and the time delays of all the species in reference to protons are listed in table I.

Table I. The velocities and time delays of ions arriving at the target for different ion species. The time for proton arriving at the target is refereed as zero point in time space.

Ion species	H ¹⁺	C ⁴⁺	C ³⁺	C ²⁺	C ¹⁺
Velocities(cm/ns)	2.5	0.8	0.6	0.4	0.2
Time delays in relative to H ¹⁺ (ns)	0	29.8	44.3	73.5	161.0

Remark 3: *2) that the highly charged target C and O ions do not contribute to the MeV proton stopping despites that proton velocity is above the target thermal electron one.*

Response and revision: The energy loss of intense proton beam in ionized matter is composed of two terms as $dE/dx = (dE/dx)_{collision} + (dE/dx)_{collective}$. The first term $(dE/dx)_{collision}$ describes the collisional stopping induced by binary interaction of the individual projectiles with the individual particles in the plasma. The second term $(dE/dx)_{collective}$ describes the collective stopping induced by the beam-driven electric fields.

$(dE/dx)_{collision}$ consists of contributions from free electrons and plasma ions. After heating, the average ionization degree for the current CHO target is about $C^{3.8+}H^{0.98+}O^{4.5+}$. Therefore the contributions from plasma ions include two parts, bound electrons and nuclei.

In the current regime, where the protons are much faster than the thermal electrons, the contribution of the nuclei to $(dE/dx)_{collision}$ can be neglected [T. Peter and J. Meyer-ter-Vehn, Phys. Rev. A 43, 1998 (1991)]. Here in this article the nuclear stopping is excluded, the contribution of bound electrons are included in the calculations.

In the revised manuscript, this is underlined in the first paragraph of discussion section.

Remark 4: *IF the authors could achieve that goal, they would add a splendid contribution to the very challenging field of proton stopping in dense classical plasma.*

Response: We again thank the referee for appreciating our work. An accurate understanding of the beam transportation behavior in dense matter is crucial for all these applications. We report the first experimental evidence on one order of magnitude enhancement of intense laser-accelerated proton beam stopping in dense ionized matter, in comparison with the current-widely used models describing individual ion stopping in matter. To ensure correct interpretation of the measurement, the energy spread of the intense proton beam was remarkably reduced by a bending magnet, and the dense ionized matter was homogeneously created by heating a foam with soft X-rays from a gold hohlraum. Supported by Particle In Cell (PIC) simulation, we attribute the enhancement to the strong decelerating electric field approaching 1 GV/m that can be created by the beam-driven return-current. This collective effect plays the dominant role in the stopping of laser-accelerated intense proton beam in dense ionized matter. This finding is essential for the optimum design of ion driven fast ignition and inertial confinement fusion.

Reply to the Referee #2:

Remark 1: *Major comments and report summary.*

In their manuscript "Anomalous stopping of laser-accelerated intense proton beam in dense ionized matter" the authors present results both collected in a experiment and modelling using the PIC simulation technique on stopping of protons in an isochorically heated target medium. I do not recommend publication of this manuscript in its current form for two major reasons given in the following.

While the presented scientific results on stopping power seem sound and are indeed interesting.

Response: We thank the referee for appreciating our work.

Remark 2: *(1) the style of presentation is poor across the manuscript (see comments and highlights in the annotated PDF)*

Response: We made thorough revisions according to the reviewer's comments (see details in the response of remark 4-15 below). Additional errors in grammar and typos, that are highlighted in the annotated PDF, are corrected as well. We really appreciate the referee's constructive comments, which helped us to significantly improve our manuscript both in language and logic.

Remark 3: *(2) I am not convinced the theoretical methods used to compare against experiment are sufficiently accurate in order to interpret the observed enhancement in stopping as "anomalous". In order to increase the impact of their current work, a supplemental theoretical treatment within a more accurate method (such as time-dependent density functional theory coupled to Ehrenfest dynamics) would be required to warrant publication as an article Nature Communications. The current scientific results, however, would be suitable for publication as an article in Scientific Reports or a similar journal.*

Response: Here we interpret the observed enhancement in stopping as "anomalous", because the observed energy loss is one order of magnitude higher than the predictions by the commonly-used theories such as Li-Petrasso, SSM, and Bethe-Bloch models. These models, including TDDFT (time-dependent density functional theory coupled to Ehrenfest dynamics), provide clear description for the individual ion-plasma interaction and has been demonstrated to be valid in certain conditions by a lot of experiments [just list a few here: (1)W Cayzac et al., Nature communications 8,15693(2017);(2) AB Zylstra et al., Physical review letters 114, 215002(2015);(3) JA Frenje et al., Physical review letters 122, 015002 (2019); (4)Y. H. Ding et al., Phys. Rev. Lett. 121,145001(2018)]. However, the situation is quite different in our experiment.

We modelled the interaction with PIC simulation, which contains both close collisions and collective electromagnetic fields. It is found that strong decelerating electric field approaching 1 GV/m can be created by the beam-driven return-current, and plays the dominant role in the intense proton beam stopping in dense ionized matter.

As for the TDDFT simulation, we agree with the referee that TDDFT could be more accurate. We did some primitive simulations together with Prof. Wei Kang from Peking University. After nearly two-month effort, we found that it would pose a grand challenge to the current TDDFT methodology. There are two major difficulties which need to be considered seriously. One is the difficulty of high temperature. In regular TDDFT tools provided by the open source or commercial package, the time-dependent Kohn-Sham equation is solved with electronic

wave functions. The number of electron wave function increases roughly following the rule of T^3 , with T the temperature of warm dense materials. The plasma temperature of 17 eV in experiments is beyond the computational capacity of those TDDFT package. A state-of-the-art solution to the high temperature difficulty is recently provided by Ding et al. [Y.H. Ding et al., Phys. Rev. Lett. 121, 145001(2018)], employing the orbital-free TDDFT method. Yet another difficulty is the low density of 2 mg/cm^3 of the plasma. It means not only a larger simulation size but also a larger fluctuation in the stopping power calculation caused by small number of ions in the simulation box, which requires much longer simulation time to smooth out.

In all, here in this article, we report the first experimental evidence on one order of magnitude enhancement of intense proton beam stopping in dense ionized matter, in comparison with the current-widely used models describing individual ion stopping in matter. We call this high degree of enhancement as "anomalous". The state-of-the-art PIC simulation provides a reasonable interpretation that the return-current-induced strong decelerating fields dominate the stopping process. If the term "anomalous" means something else already, we are ready to change it to more specific words, i.e. "one order of magnitude enhancement of stopping", or "unusually high degree of stopping", which is provided by the referee in remark 12.

Remark 4: *Summary of detailed comments (given in the annotated PDF)*

Page 1, left: the highest beam intensity from accelerators Style: what do you mean with "the highest beam intensity"? Elaborate.

Response and revision: In ion beam driven inertial confinement fusion (ICF) and high energy density (HED) science, extremely high-intensity ion beams are required. The higher the beam density is, the more possibilities there are to generate HED matter and achieve ICF.

In the revised version, the sentence is modified to "Besides, in the cases of ion beam driven inertial confinement fusion and high energy density science, which require extremely high beam intensity from accelerators [32-36], no collective effects on ion stopping processes due to high beam intensity are considered nor - to the best of our knowledge - were they reported in any previous experiments"

Remark 5: *Page 1, right: We ever carried out an experiment in previous, and observed a significant enhancement of energy loss for the laser-accelerated intense proton beam (see [58] and supplementary material for details). Style: this sentence is very unclear, rewrite.*

Response and revision: It is revised to "In our previous experiment, we sent the laser accelerated ion beams directly into the plasma target, and observed that the energy spectra of the ions were significantly downshifted after passing through the dense plasma. This energy downshift was far beyond the Bethe-Bloch predictions (see [64] and supplementary material for details). "

Remark 6: *Page 2, left: However, it was difficult to conduct quantitative analysis due to the large energy spread of the beam. In order to improve our understanding of these effects, experimental data with high precision are required. Style: unclear, rewrite. I believe, the motivation of this work is a previously conducted experiment Ref. 58. Please rewrite this paragraph accordingly.*

Response and revision: We rewrote this paragraph to "In our previous experiment, we sent the laser accelerated ion beams directly into the plasma target, and observed that the energy spectra of the ions were significantly downshifted after passing through the dense plasma. This

energy downshift was far beyond the Bethe-Bloch predictions (see [64] and supplementary material for details). However, the large energy spread of the incident beam makes it difficult to correctly interpret the results. In this article, we improve the precision of the measurement by using a magnetic dipole to trim out a quasi-mono-energetic proton beam."

Remark 7: *Page 2, left: The hydrodynamic timescale of the target is long compared to the proton beam pulse, hence the target can be considered to be quasi static with well-characterized parameters. please be more quantitative: give explicit estimates for the timescales in order to justify a quasi-static approximation.*

Response and revision: It is revised to "The dense ionized target is produced by irradiating a Tri-Cellulose Acetate (TCA) foam sample with soft X-rays from a laser-heated hohlraum. Thus the temperature and density are homogeneous across the ionized target. This state can be maintained for a period of more than 10 nanoseconds. This period is two to three orders of magnitude longer than the beam duration and the beam-plasma interaction timescale. Therefore, the target can be considered to be quasi-static."

Remark 8: *Page 2, left: We observed enhanced stopping by one order of magnitude compared to the classical single particle slowing down theory. This seems to be the central result of this work. However, the style of presentation in this paragraph is very poor. Also, a classical single-particle theory is used to compare against the experimental result. However, it is not clear at all which theory the authors compare against. Do you compare against (1) Bethe-Bloch model, (2) Li-Petrasso (LP) theory or something else? Please clarify and add citations accordingly.*

Response and revision: We added the citation accordingly. The sentence is revised to "We observed enhanced stopping by one order of magnitude (see Fig. fig3) in comparison with the predictions from individual ion stopping theories, including Bethe-Bloch[45,46], Li-Petrasso (LP)[43], Standard Stopping Model (SSM)[49]. "

Remark 9: *Page 2, left: We attribute this phenomenon to a strong decelerating electric field induced by the intense proton beam. Refer to results and figures here.*

Response and revision: The result and figures are referred. The sentence is revised to "Through PIC simulation, we attribute the high degree of enhancement to a strong decelerating electric field induced by the intense proton beam (see Fig. 4)."

Remark 10: *Page 3, right: Due to the large penetration of soft x-rays, the foam was heated quasi isochorically. Clarify: isochorically or quasi-isochorically? If quasi-isochorically, please explain.*

Response and revision: The foam target has a sponge-like structure with pore size of $\sim \mu\text{m}$. Once heated by soft X-rays from hohlraum radiation, the material expansion occurs inside the target between the sponge-like structures. This micro expansion leads to target homogenization, while the volume and density of the entire target stay constant for more than 10 nanoseconds. This was demonstrated in our previous experiment (Ref. 66 in the manuscript).

In the revised manuscript, this is clarified as " The X-rays subsequently irradiated and heated the foam target ($\text{C}_9\text{H}_{16}\text{O}_8$, density of 2 mg/cm^{-3} and thickness of 1 mm) to produce ionized

matter. The hydrodynamic response of the heated foam target under this kind of scheme has been very well investigated and the state has been well characterized [65-67]. Once heated by soft X-rays from hohlraum radiation, the material expansion occurs inside the target between the sponge-like structures. This micro expansion leads to target homogenization, while the volume and density of the entire target stay constant for more than 10 nanoseconds. Therefore, homogeneous, ns-long living, and quasi-static ionized matter is generated.

Remark 11: *Page 3, right: In Fig. 3, the measured energy loss is compared to theoretical models, e.g. Bethe-Bloch model, Li-Petrasso (LP) theory [38] and Standard Stopping Model (SSM) by Deutsch [44]. These theories are based on binary collisions with free electrons, bound electrons and/or plasmons. They all underestimate the measured stopping power by as much as one order of magnitude. We therefore call the observed effect anomalous stopping and attribute this to collective electromagnetic effects induced by high-current ion beams. Fair enough, but these models are very simple. In my opinion, a comparison with very simplified theoretical models is not sufficient to conclude that the experimentally observed stopping is anomalous. A comparison with a higher level of theory is required. The observed result is still interesting, but I believe, a more thorough theoretical analysis is required to attribute an "anomaly" to the observed stopping. I would suggest to compare against results computed within time-dependent density functional theory coupled to Ehrenfest dynamics which is the current "gold standard". These calculations will be computationally expensive but doable under the reported conditions (a temperature of 17 eV, mass density 2 mg/cc, MeV velocity of projectile).*

Response and revision: As mentioned before, we call the observed enhancement in stopping as "anomalous", because the observed energy loss is one order of magnitude higher than the predictions by the commonly-used individual ion stopping theories. According to the referee's comments, we did some primitive simulations together with Prof. Wei Kang from Peking University. After two-month effort, we found it would pose a grand challenge to the current TDDFT methodology. However, the state-of-the-art PIC simulation provides a reasonable interpretation that the return-current-induced strong decelerating fields dominate the stopping process.

In the revised manuscript, we thank Prof. Kang's team for performing TDDFT calculations in the "acknowledge" section. The publication related to TDDFT was cited.

Remark 12: *Page 3, right: In order to understand this anomalous stopping, both collective electromagnetic effects and close particle... Bad style of presentation: Only here is becomes clear that the authors actually perform additional modeling in order to find an interpretation of the unusually high degree of stopping observed. This was not mentioned at all up to this point. I recommend the authors rewrite the abstract and their introduction and clearly explain their methodology, particularly with regard to the PIC simulations.*

Response and revision: We rewrote the abstract and introduction sections. The PIC methodology is also concisely and clearly described in the discussion section and method section, respectively. Please see details in the text. Thanks to the referee's comments, the whole presentation style is greatly improved.

Remark 13: *Page 4, left, Fig. 3: Can you also give error estimates for the models (Bethe-Bloch,*

Li-Petrasso, SSM)?

Response and revision: The error of these analytical models is estimated as about 15%, mainly originating from the uncertainty of the plasma electron density. The error estimation is added in the first paragraph of discussion section. The error bars in Fig.3 represent the full width half maximum of the respective spectra. This is precisely described in the text and the caption of Fig. 3.

Remark 14: *Page 4, left: The most appropriate tool to simulate the conditions of the experiment is the Particle-in-Cell method (PIC), which in recent years has established itself as a state-of-the-art method for solving problems of kinetic plasma physics. Add citation(s) to seminal paper(s) and point to supplemental material for details on the theory used.*

Response and revision: The seminal papers related to PIC models are cited. Details of the theory are referred to method section. It is revised to "The most appropriate tool to simulate the conditions of the experiment is the Particle-in-Cell method (PIC) (see details in Method), which in recent years has established itself as a state-of-the-art method for solving problems of kinetic plasma physics [54,76,77]. "

Remark 15: *Page 5, right: We used the newly developed PIC code LAPINS [65, 66], where... The authors should have referenced these citations already when they discuss their PIC calculations. Although references are given, the authors should also provide a concise and clear description of their PIC methodology either here or in the supplementary material.*

Response and revision: The citations about the LAPINS code are upfronted to the second paragraph of the discussion section, and a concise description is added as well. More clear description of PIC methodology is added in the method section. See details in the text.

Reply to the Referee #3:

Remark 1: *Comments on "Anomalous stopping of laser-accelerated intense proton beam in dense ionized matter" written by Jieru Ren et al., to be submitted to Nature Communications. In this paper, the authors mention experimental results of anomalous stopping power of MeV ion beam in dense ionized matter. They use laser generated ion beam with magnetic dipole energy filter and radiation heated target with high energy laser pulse.*

The method itself looks interesting and their idea to generate quasi-monochromatic ion beam from very broad TNSA ion beam looks well organized operation.

Response: We thank the referee for appreciating our work.

Remark 2: *They claim that they measured anomalous stopping phenomena with energy shifted ion beam energy and comparison with the PIC simulation prediction. The reviewer carefully read this manuscript and it is several difficulties to understand the details of the experiments and to judge their claims. The reviewer lists up the uncertain points and considers the authors should make them clear in their manuscript.*

(1) There is no information about the current of quasi-monochromatic ion beam. That ion beam is sliced with single magnetic dipole filter. Therefore, the current itself largely decreases from generation point. In addition, energy angle dispersion of ion beam remains after passing through the dipole magnet so that cross section of ion beams gradually increases after the slit of the magnetic dipole filter. In that situation, how much current density can they expect and is that number still enough high to use the quasi-monochromatic ion beam for such scientific investigation? They need to show it.

Response: Two CR39 detectors are mounted at the positions of the target and the TPS respectively to record the beam profile. The half-divergence angle of the selected beam is examined to be 4.4 mrad after passing through the dipole. The experimental evidence is shown in Fig. 1 of the supplementary materials. The CR39 detectors saturated due to the high current. A rough estimation of beam density reduction by 5 orders (to $10^{14-15} \text{ cm}^{-3}$) from the source to target is underlined in the supplementary material. We point out this in the supplementary material as well. Hence after the dipole filtering, the beam density is still several orders of magnitude higher than that can be achieved by large-scale modern accelerators like FAIR (facility for antiproton and ion research). The obtained high-intensity quasi-monoenergetic ion beams provide great opportunities for studies of intense ion beam stopping and transportation process in matter.

Remark 3: *(2) In such energy loss investigation, comparison between perturbed and unperturbed spectrum is essential part. However, there is almost no information about how to get this unperturbed ion spectrum. They only said 'without target'. This means nothing between quasi-monochromatic filter and the spectrometer. If that so, it is some difficulties to use these different transport circumstance conditions as reference. If they have heated and unheated target results, that would be more reliable.*

Response: We measured the ion spectra under three cases, namely there are no target (without target), or an unheated target (cold target), or a heated target (plasma) between the quasi-monochromatic filter and the Thomson parabola spectrometer. The results about the unheated target are shown in the supplementary material. In this article, we focused on the discussion

about the energy loss in the heated target. To obtain the energy loss data, we analyze the downshift of the energy spectra between the case of "heated target" and the case of "no target" .

Remark 4: (3) *In many TNSA ion beam probe experiments, there are transit potential effect of the high current ion beam illuminated target. There are many distorted mesh pictures from the ion beam deflection but there are negligible small electrostatic potential effect of the target. (for example, Mackinnon et al., Rev. Sci. Instrum., vol.75, No.10, p.3531 (2004))They need to mention about their loosing of the energy purely occurs inside the heated target.*

Response and revision: The TNSA proton beam has been used in many experiments to probe the electromagnetic fields in laser-produced plasmas. Parts of the work are listed below, including the reference mentioned by the referee. In those work, the thin foil target is heated directly by the high-power lasers (typically 10^{15} W/cm² or above), and the probing proton beam is sent directly to the laser-foil interaction region (typically hundreds of μm or less) within the interaction period (typically hundreds of picoseconds).

However, in our case, the ionized matter was generated through indirect heating scheme. Namely, a gold hohlraum converter was used to generate soft x-ray radiation by interaction of a ns laser pulse with the hohlraum walls. The X rays subsequently irradiated and heated the foam target to the ionized state. In this way, 1) the proton beam is 1 mm away from the focal spot of the ns laser; 2) the proton beam reaches the target at about 10 ns after the ns-laser pulse; 3) the target is heated by the soft X rays from the hohlraum at an irradiance of about 10^{13} W/cm². Therefore, the fields caused by the heating process are negligible, and the protons lose the energy purely inside the heated target.

In the revision version, we added "Unlike producing ionized matter through direct heating by high-power lasers - in which case strong electromagnetic fields are generated in the target and will greatly influence the proton transportation behavior [68-70] - here the electromagnetic fields resulted from the heating process can be neglected."

1) G. Sarri, A. Macchi, C. A. Cecchetti, S. Kar, T.V. Liseykina, X. H. Yang, M. E. Dieckmann, J. Fuchs, M. Galimberti, L. A. Gizzi, R. Jung, I. Kourakis, J. Osterholz, F. Pegoraro, A. P. L. Robinson, L. Romagnani, O. Willi, and M. Borghesi, Dynamics of Self-Generated, Large Amplitude Magnetic Fields Following High-Intensity Laser Matter Interaction, Phys. Rev. Lett. 109, 205002 (2012);

2) A. J. Mackinnon, P. K. Patel, R. P. Town, M. J. Edwards, T. Phillips, S. C. Lerner, D. W. Price, D. Hicks, M. H. Key, S. Hatchett, and S. C. Wilks, M. Borghesi, L. Romagnani, and S. Kar, T. Toncian, G. Pretzler, and O. Willi, M. Koenig, E. Martinolli, S. Lepape, A. Benuzzi-Mounaix, and P. Audebert, J. C. Gauthier, J. King, R. Snavely, and R. R. Freeman, T. Boehlly, Proton radiography as an electromagnetic field and density perturbation diagnostic, Rev. Sci. Instrum. 75, 3531 (2004);

3) M. Borghesi, L. Romagnani, A. Schiavi, D. H. Campbell, M. G. Haines, O. Willi, A. J. Mackinnon, M. Galimberti, L. Gizzi, R. J. Clarke, and S. Hawkes, Measurement of highly transient electrical charging following high-intensity laser-solid interaction, Applied Physics Letters 82, 1529 (2003);

4) L. Romagnani, M. Borghesi, C. A. Cecchetti, S. Kar, P. Antici, P. Audebert, S. Bandhoupad-jay, F. Ceccherini, T. Cowan, J. Fuchs, M. Galimberti, L. A. Gizzi, T. Grismayer, R. Heathcote,

R. Jung, T. V. Liseykina, A. Macchi, P. Mora, D. Neely, M. Notley, J. Osterholtz, C. A. Pipahl, G. Pretzler, A. Schiavi, G. Schurtz, T. Toncian, P. A. Wilson and O. Willi, Proton probing measurement of electric and magnetic fields generated by ns and ps laser-matter interactions[J]. Laser and Particle Beams, 2008, 26(2): 241-248.

Remark 5: (4) Normally, Thomson parabola spectrometer needs pinhole entrance pinhole to define the starting point before deflection of electric field and magnetic field. But, this pinhole play a role to produce additional potential effect if the ion current density is so high. In their experimental setup, there is no information about it. In addition, in Fig. 2(a), there is relatively broad 'zero point' signal. It may be neutral components of charge exchange process. But that narrowness of neutral component in the Thomson spectrometer is one of the experimental evidence of the energy resolution. It looks large width and spreading in the only B field direction. To clarify the energy difference in their experiments, they need to clarify their spectrometer condition and resolution more seriously.

Response and revision: If we understand the comment about the potential effect of the pinhole correctly. The reviewer means that the intense ion beam leads to ionization or charging of the pinhole material, which generates additional potential. We think this effect can be neglected for the following reasons: 1) CR39 plastic detector is stacked in front of the pinhole, therefore, most of the particles are stopped in the plastic; 2) The pinhole is grounded.

It is really a good question referring to the zero point. The zero point in Fig. 2 of the manuscript is formed by the neutral components of the charge exchange process. We scanned the TPS pinhole (see Fig. 1 below), and the image agrees perfectly with the measured distribution of zero point. Therefore, the large width and spreading of the beam in the only B field direction is caused by the similar shape of the pinhole.

The spectrometer condition and resolution are added in the text as " The pinhole size of the TPS is $180\ \mu\text{m} \times 135\ \mu\text{m}$ in magnetic and electric deflection direction respectively, and the energy resolution of protons at 1 MeV achieves $E/\delta E=17$ in B deflection direction. More information about the TPS can be found in Ref. [72]"

Fig. 1. Scanned image of the entrance hole of TPS.

Remark 6: (5) Actually, there are many papers related to the anomalous stopping of ion

beams. They need to add more references. Specially, they miss old Sandia's ion beam fusion experiments. (for example, F. C. Young, PRL, Vol. 49, No.8, p.549 (1982), E. J. McGuire et al., Phys. Rev. A, Vol. 26 No.3, p.1318 (1982))

Response and revision: The works mentioned by the referee are really important and pioneering. The co-authors have worked on the stopping of ion beams in ionized matter for about 20 years. The papers from Young and McGuire et al. are often used. In the revised version, the mentioned references are added, together with more references from Hoffmann, Jacoby and Rosmej et al.

Remark 7: (6) *In the previous accelerated ion beam experiment, there are ionizing effect of the target is one of the mechanism to explain enhancement stopping power. In the author's experimental condition, target carbon atoms are not fully ionized. Therefore, they need to include this effect.*

Response and revision: Yes. In previous energy loss experiments, it was demonstrated that the energy loss of protons in ionized matter is higher than in neutral/cold matter, because of the difference in Coulomb logarithms. The physical reason is that free electrons are more easily excited by the beam ions (plasma waves) than bound electrons (bound-bound excitations and bound-free ionizations). In fact, for very intense ion beams, the energy loss is composed of two terms as $dE/dx = (dE/dx)_{collision} + (dE/dx)_{collective}$. The first term $(dE/dx)_{collision}$ describes the collisional stopping induced by binary interaction of the individual projectiles with the individual particles in the plasma such as plasma free electrons, plasma bound electrons, and plasma nuclei. The second term $(dE/dx)_{collective}$ describes the collective stopping induced by the beam-driven electric fields.

$(dE/dx)_{collision}$ consists of contributions from free electrons and plasma ions. Since the current target is partially ionized, the contributions from plasma ions include two parts, bound electrons and nuclei.

In the current regime, where the protons are much faster than the thermal electrons, the contribution of the nuclei to $(dE/dx)_{collision}$ can be neglected [T. Peter and J. Meyer-ter-Vehn, Phys. Rev. A 43, 1998 (1991)]. Here in this article the nuclear stopping is excluded. The contribution of free electrons and bound electrons are studied in the article.

Remark 8: (7) *The author's main claiming points is induced electric field affecting to the stopping power of ion beams. If they want to appeal this, the reviewer consider the comparison between low current density experiment and high current density experiment. But they have no data about that. In this case, it is very difficult to say their mechanism is dominated process. If they have some other experimental data related and can be used for explanation of their model, they need to show them.*

Response and revision: Coordinated by Dr. Rosmej, we carried out the low-current-density experiments at GSI using 4.77 MeV/u Ti ions from UNILAC and laser from PHELIX. The target is the same as the current one. The incident ion energies are in the same regime where projectiles are much faster than the thermal electrons. The experimental setup is shown in Fig. 2 (a) here. The ion energies were analysed by means of Time of flight (TOF) method. The typical TOF spectra of the ions in cases of no target, cold target and heated target are shown in Fig. 2 (b) here. It is found that the measured energy loss of about 8 MeV/mm fairly well agrees

with the individual ion stopping theory (Bethe-Bloch) prediction 8.6 MeV/mm (Ref. 42 in the manuscript). In the revised manuscript, the low-current-density results are cited.

Fig. 2. (a) Experimental setup of measuring low-current ions stopping in dense ionized matter; (b) Typical TOF spectra of ions in cases of no target, cold target and heated target. Figures are from Ref. 42 in the manuscript.

We hope we have addressed all comments of the Referees, and consequently would like to ask for consideration of publication in Nature Communications.

Yours sincerely,

Jieru Ren, Zhigang Deng, Wei Qi, Benzheng Chen, Bubo Ma, Xing Wang, Shuai Yin, Jianhua Feng, Wei Liu, Zhongfeng Xu, Dieter H.H. Hoffmann, Shaoyi Wang, Quanping Fan, Bo Cui, Shukai He, Zhurong Cao, Zongqing Zhao, Leifeng Cao, Yuqiu Gu, Shaoping Zhu, Rui Cheng, Xianming Zhou, Guoqing Xiao, Hongwei Zhao, Yihang Zhang, Zhe Zhang, Yutong Li, Dong Wu, Weimin Zhou, and Yongtao Zhao

Reviewer #1 (Remarks to the Author):

Lots of Thanks for overcoming every referee hurdle.
You have successfully completed the experimental proposal initiated in Ref.55 through the hohlraum target monitoring.

Reviewer #2 (Remarks to the Author):

I am glad to be able to review the revised version of the manuscript "Anomalous stopping of laser-accelerated intense proton beam in dense ionized matter".

I have rarely seen such an improvement on a previously submitted manuscript. According to my suggestions, the authors have revised and improved their manuscript substantially.

I am also impressed by their attempt to perform TDDFT simulations as I had suggested. I disagree with the authors that the temperature (17 eV) is a limiting issue for performing these simulations, but I agree that the low mass density will make TDDFT calculations computationally infeasible.

My only remaining concern is the use of the term "anomalous". As discussed in the correspondence with the authors, the current comparison with simpler theories (Bethe-Bloch, Li-Petrasso, SSM, and even PIC) does not warrant to call the observed effect as "anomalous". Instead, I suggest to call this observation an "unusually high degree of stopping" (or something along those lines) as the authors suggested in their response. An in-depth analysis on the microscopic level by means of a first-principles modeling technique (which currently seems computationally infeasible) would help elucidate this issue.

Additional, minor comments on typos are given in the annotated pdf.

I now recommend publication in Nature Communications, once the author have reconsidered my request above.

Reviewer #3 (Remarks to the Author):

Comments on "Anomalous stopping of laser-accelerated intense proton beam in dense ionized matter" written by J. Ren, et al., to be submitted to Nature Communications. The authors show new type of ion beam stopping experiments by using laser accelerated intense proton beam and radiation heated matter. When the reviewer read the reply comments to the reviewer, almost all the responses are clear. However, about the energy resolution point, it is not consistent explanation.

They said pinhole size decide resolution and the corresponding energy resolution is $E/\delta E \sim 17$ at 1MeV. This value is almost proportional to square root of the mean energy of ions, so that at $E = 3.36\text{MeV}$, $\delta E \sim 0.36\text{ MeV}$. But, in Fig.2, they achieve more small energy spread of ion beam without target condition. In addition, the previous comments from Reviewer #3 said, the spread area of the zero point (the track of the neutral atoms) denotes more wider to compare the above energy spread. Therefore, the authors should clarify (1) why the neutral zero point has so large? (2) How do they decide the energy resolution of their spectrometer? (3) What effect of this energy resolution to their conclusion?

The energy difference shown here between without plasma and after plasma is 0.38MeV. Normally, energy resolution should be enough smaller than this value.

Reply to the Referee #1:

Remark 1: *Lots of Thanks for overcoming every referee hurdle. You have successfully completed the experimental proposal initiated in Ref.55.through the hohlraum target monitoring.*

Response: We thank the Referee for appreciating our work.

Reply to the Referee #2:

Remark 1: *I am glad to be able to review the revised version of the manuscript "Anomalous stopping of laser-accelerated intense proton beam in dense ionized matter". I have rarely seen such an improvement on a previously submitted manuscript. According to my suggestions, the authors have revised and improved their manuscript substantially.*

Response: We thank the Referee for appreciating our work.

Remark 2: *I am also impressed by their attempt to perform TDDFT simulations as I had suggested. I disagree with the authors that the temperature (17 eV) is a limiting issue for performing these simulations, but I agree that the low mass density will make TDDFT calculations computationally infeasible.*

Response: We agree with the Referee that the temperature is indeed not a limiting issue in TDDFT calculations. For instance, Ding et al. [Phys. Rev. Lett. 121, 145001(2018)] have studied the stopping power of tens-of-eV solid-density plasmas with time-dependent orbital-free density functional theory (TD-OF-DFT) method.

Remark 3: *My only remaining concern is the use of the term "anomalous". As discussed in the correspondence with the authors, the current comparison with simpler theories (Bethe-Bloch, Li-Petrasso, SSM, and even PIC) does not warrant to call the observed effect as "anomalous". Instead, I suggest to call this observation an "unusually high degree of stopping" (or something along those lines) as the authors suggested in their response. An in-depth analysis on the microscopic level by means of a first-principles modeling technique (which currently seems computationally infeasible) would help elucidate this issue.*

Response and revision: According to the Referee's suggestion, we have changed the words "anomalous" in the revised manuscript. Please see details in the text.

Remark 4: *Additional, minor comments on typos are given in the annotated pdf.*

Response and revision: The typos have been corrected according to the comments. Please see details in the text.

Remark 5: *I now recommend publication in Nature Communications, once the author have reconsidered my request above.*

Response: Thanks. We have revised our manuscript according to the Referee's comments.

Reply to the Referee #3:

Remark 1: *Comments on "Anomalous stopping of laser-accelerated intense proton beam in dense ionized matter" written by J. Ren et al., to be submitted to Nature Communications. The authors show new type of ion beam stopping experiments by using laser accelerated intense proton beam and radiation heated matter. When the reviewer read the reply comments to the reviewer, almost all the responses are clear.*

Response: We thank the Referee for appreciating our work.

Remark 2: *However, about the energy resolution point, it is not consistent explanation. They said pinhole size decide resolution and the corresponding energy resolution is $E/\delta E \sim 17$ at 1 MeV. This value is almost proportional to square root of the mean energy of ions, so that at $E = 3.36$ MeV, $\delta E \sim 0.36$ MeV. But, in Fig.2, they achieve more small energy spread of ion beam without target condition. In addition, the previous comments from Reviewer #3 said, the spread area of the zero point (the track of the neutral atoms) denotes more wider to compare the above energy spread. Therefore, the authors should clarify*

(1) why the neutral zero point has so large ?

(2) How do they decide the energy resolution of their spectrometer?

(3) What effect of this energy resolution to their conclusion ? The energy difference shown here between without plasma and after plasma is 0.38 MeV. Normally, energy resolution should be enough smaller than this value.

Response and revision: We thank the Referee for the helpful comments. Please see our point-by-point response below.

(1) The particles at the zero point originate from the neutralization of selected ions, i.e. protons and carbon ions. The size of the zero point is therefore comparable with the size of those ions. As shown in Fig. 2(a) of the manuscript, the C^{1+} ions have a quite wide distribution, which may contribute to a "large" zero point.

(2) In principle, the energy resolution of the TPS at a specific energy E is characterized by $E/\delta E$, where δE is the energy range covered by the incident beam spot on the detector. In the nonrelativistic case, $E/\delta E \approx y/2s$, where y is the deflection distance proportional to the square root of the mean energy and s is the beam spot size on the detector depending on the pinhole size and the distances from the ion source to the pinhole and the detector plane [Jung et al., Rev. Sci. Instrum. 82,013306(2011); Rajeev et al., Rev. Sci. Instrum. 82,083303(2011)].

Our measurement shows that the selected quasi-monoenergetic proton beam has the spot size (FWHM) of $140 \mu\text{m}$ (see Fig. 1 below). This corresponds to δE of 0.10 MeV and the energy resolution of $E/\delta E \sim 34$ at 3.36 MeV. What's more, the energy resolution is calculated via the equation $E/\delta E \approx y/2s$, where y is measured to be 9.5 mm for 3.36 MeV protons, and s is measured to be $140 \mu\text{m}$ in B deflection direction. In this way, the same energy resolution of $E/\delta E \sim 34$ is obtained.

In the previous manuscript, the energy resolution, $E/\delta E \sim 17$ at 1 MeV for protons, was greatly under-estimated, since it was simply linearly-scaled by the pinhole size in reference to the results given in Ref.[72], where the beam spot size is much larger than that in the current case.

In the revised manuscript, the energy resolution for protons is corrected to be $E/\delta E \sim 34$ at 3.36 MeV. The detailed calculations are complemented in the supplementary material.

Thanks for the Referee's comment, the authors are aware of a typo in the previous manuscript: The measured proton energy spectra were fitted by the Gaussian function $f(x)=a \times \exp(-(x-b)^2/c^2)$, where $c = \sqrt{2}\sigma \approx \text{FWHM}/1.67$, and σ is the variance. We wrongly took the value of c as the FWHM, which has been corrected in the revised manuscript.

Fig. 1. The spatial distribution of the proton tracks in B deflection direction recorded on TPS CR39 without target.

(3) The energy resolution of the current measurement has been corrected to be $\delta E \sim 0.10$ MeV at 3.36 MeV, which is enough to distinguish the energy shift of 0.38 MeV between "without plasma" and "after plasma" cases.

We again thank the Referees for the helpful comments. We hope we have addressed all the comments.

Yours sincerely,

Jieru Ren, Zhigang Deng, Wei Qi, Benzheng Chen, Bubo Ma, Xing Wang, Shuai Yin, Jianhua Feng, Wei Liu, Zhongfeng Xu, Dieter H.H. Hoffmann, Shaoyi Wang, Quanping Fan, Bo Cui, Shukai He, Zhurong Cao, Zongqing Zhao, Leifeng Cao, Yuqiu Gu, Shaoping Zhu, Rui Cheng, Xianming Zhou, Guoqing Xiao, Hongwei Zhao, Yihang Zhang, Zhe Zhang, Yutong Li, Dong Wu, Weimin Zhou, and Yongtao Zhao

Reviewer #3 (Remarks to the Author):

The reviewer agrees with the acceptance of this paper in Nature Communications. After revising with several comments from referees, this manuscript clearly denotes the point the authors want to appeal. This experiment, itself, is very difficult and its analyzation is not so easy. However, they can show some clear difference between plasma and non-plasma condition. Even though, there is some uncertain to decide the details in physics, but results of experimental research works with such ultra-high power lasers, it is enough to be published. We also expect the readers of this article will open mind to feel the next generated experiments with ultra-high power lasers. It is very important.

Reply to the Referee #3:

Remark 1: *The reviewer agrees with the acceptance of this paper in Nature Communications. After revising with several comments from referees, this manuscript clearly denotes the point the authors want to appeal. This experiment, itself, is very difficult and its analyzation is not so easy. However, they can show some clear difference between plasma and non-plasma condition. Even though, there is some uncertain to decide the details in physics, but results of experimental research works with such ultra-high power lasers, it is enough to be published. We also expect the readers of this article will open mind to feel the next generated experiments with ultra-high power lasers. It is very important.*

Response: We thank the Referee for appreciating our work.

Yours sincerely,

Jieru Ren, Zhigang Deng, Wei Qi, Benzhen Chen, Bubo Ma, Xing Wang, Shuai Yin, Jianhua Feng, Wei Liu, Zhongfeng Xu, Dieter H.H. Hoffmann, Shaoyi Wang, Quanping Fan, Bo Cui, Shukai He, Zhurong Cao, Zongqing Zhao, Leifeng Cao, Yuqiu Gu, Shaoping Zhu, Rui Cheng, Xianming Zhou, Guoqing Xiao, Hongwei Zhao, Yihang Zhang, Zhe Zhang, Yutong Li, Dong Wu, Weimin Zhou, and Yongtao Zhao